# Linking Off-Road Points to Routing Networks

**Dominik Köppl** 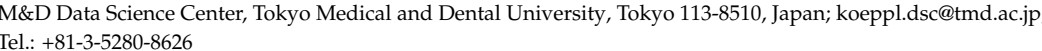

M&D Data Science Center, Tokyo Medical and Dental University, Tokyo 113-8510, Japan; koeppl.dsc@tmd.ac.jp; Tel.: +81-3-5280-8626

**Abstract:** Although graph theory has already been introduced in spatial reasoning, current spatial database systems do not provide out-of-the-box routing on geometric points that are not matched on the graph. Methods that connect new reference locations to the graph render different routing results. Moreover, current solutions break reasoning down to local analysis. We bridge the gap between routing networks and spatial geometry by a global matching of geometric points to routing networks.

**Keywords:** routing networks; modeling; spatial databases; geospatial reasoning; uncertainty

## 1. Introduction

With the growth of the number of new query dimensions that address spatial and temporal properties, business intelligence analysts claim that the buzz-word term Big Data [1] does nowadays put less emphasis on the sheer size of data but focuses more on the *v*-terms variety and veracity, which translate to the context of spatial data in the following ways.

**Variety** Spatial reference systems encode spatial data by relative location (e.g., coordinates on a local projection) or absolutely (e.g., WGS84 [2] or the Mercator projection [3]) in various formats.

**Veracity** Every dataset that provides spatial data is incomplete and inaccurate in the open-world assumption, for instance, the dynamic reshaping of infrastructure.

This can be seen by two facts: On the one hand, in most parts of the world, modern network data are approaching nearly a perfect mapping between digital representation and real-life infrastructure. On the other hand, diverse sources of spatial data emerge due to the vast availability of mobile gadgets that collect location data (e.g., geotagging, -caching, etc.). This phenomenon involves structured geoinformation data sources such as Open Street Map (OSM), as well.

Nowadays, spatial database systems have already taken the bait to provide useful query access to data with spatial information [4]. They enable fast spatial queries that may involve routing, but they do not provide the easy aggregation of diverse data formats. This comes apparently to light when dealing with routing networks and spatial objects. Common spatial systems map the network locally on a plane such that the Euclidean metric can be used for fast geographic reasoning.

On the one side, adding, moving or deleting vertices or edges can be naturally interpreted geometrically. On the other side, naïve approaches neglect the following problems:

1. First, the network's graph might not be planar. When projecting a non-planar graph to a plane, the image has intersecting lines that represent two edges. By merely looking at the image, we cannot differentiate whether it is a crossing or whether one street tunnels under the other. A geometrical query, unaware of this fact, might deliver results that do not match real-world expectation. Let us, for instance, consider a long bridge crossing a valley. A query for the nearest access point to the routing network of the valley might suggest this bridge as a result, although it is not directly accessible from the valley.

2.　A second problem arises when very distant points are taken into consideration. For large routing networks, it is not possible to project them on a compact plane that respects both the angles and distances of the Earth. That is because the Earth is not isomorphic to any compact flat model. For instance, let us take a look at the Mercator projection [3]. It gained popularity due to its accuracy of angles and is still used for course information in marine navigation systems [5]. Furthermore, the projection provides a good approximation for nearby spatial reasoning. Unfortunately, the projected shapes of the objects get distorted with respect to angle and length. More precisely, the distortion correlates with the objects' distances to the equator. Hence, the projection exaggerates the distances and sizes of spatial objects near pole regions. This makes spatial objects with large differences in latitude incomparable, and is a main complaint of the Mercator projection. In order to cope with large-distance reasoning, we have to leave the concept of a single chart and move towards a concept that represents a "truer" form of the Earth. In other words, distances between distant objects are retrieved by a geodesic line instead of some projection.

Having recognized these facts, we want to deal with the following emerging problem.

**Research Hypothesis.**　Given a (global) routing network that is based on a geometric representation, we want to enhance the network in such a way that queries based on geometric information are answered by matching the geometric position with the network. This should be conducted in such a way that respects the nature of the geometric representation.

Nowadays, most current spatial systems are based on the World Geodetic System [2]. Its underlying model shapes the Earth as a spheroid. In fact, when neglecting local height differences, e.g., mountains and valleys, it is a good approximation.

**Remark 1** ([6]). *The representation of the Earth as a spheroid can be formalized by the local parametrizations*

$$\phi_j : [0, \pi] \times I_j \to \mathbb{R}^3, \ (\theta, \psi) \mapsto \begin{pmatrix} a \ \sin\theta \ \cos\psi \\ b \ \sin\theta \ \sin\psi \\ c \ \cos\theta \end{pmatrix},$$

*with $I_1 = (0, 2\pi)$ and $I_2 = (-\pi, \pi)$, $j = 1, 2$, where the variables $a, b, c \in (0, \infty)$ have to be chosen such that the overall error is minimized [7]. Here, $\theta$ denotes the inclination (i.e., latitude) and $\psi$ the azimuth (i.e., longitude). The intervals $I_j$ are chosen in such a way that there are no degenerated points, i.e., each inverse function and its derivate are continuously defined.*

In this article, we show an approach combining a routing network with a geometrical representation that respects distances globally. In simple terms, we augment a given routing network's graph with new vertices in such a way that the new graph can give us routing information from and to the newly created vertices that reference geometrical points that were not yet represented by the routing network. To keep it short, we use the term *linkage* for such a procedure.

## 1.1. Related Work

Spatial reasoning can be conducted on graph-theoretical foundations only if all interesting points are represented by the graph. Moreover, a connected graph is necessary for useful routing queries. In most common techniques, missing parts of the street infrastructure are either:

1.　Extracted from digital imagery;
2.　Collected by sensors (e.g., GPS-enabled tracking devices);
3.　Created manually.

Every case may lead to an inaccurate or incomplete routing network. If the position is uncertain or the information about the area is incomplete, it is not possible to exactly match a location point with a vertex on a routing network. In scenario (1), the captured

images can be used to reverify the constructed routing network. Wiedemann and Ebner's algorithm [8] uses detour lengths and connection completeness as criteria when analyzing imagery. In terms of case (2), this problem is common for navigation systems that have to figure out on which path an object is moving by analyzing its trajectory. In fact, this process is so frequent that it has a name of its own—the *map matching problem* [9]. For instance, Haunert and Budig [10] used collected trajectories to discover missing road parts. Alternatively, Lou et al. [11] took an initial candidate list and use transition probability to minimize the error of choosing the right trajectory. For case (3), we have in general no additional information to ensure our decision for linking a non-mapped point to the network. The easiest setting is a planar routing network mapped on a surface: For each point of interest (POI) we want to connect, we just add an edge to the closest location of the graph. For proximity analysis, Dahlgren and Harrie [12] connected each geolocation with the nearest reference point of the routing network. de Jong and Tillema [13] took the Delaunay triangulation of their existing road network as a criterion for linking non-connected points. They further discarded those parts of the Delaunay graph that intersected with obstacles. Last but not least, Aronov et al. [14] emphasized possible detours when propagating their method to link new points to the network. They called the newly created edge a *feed-link*. Savic and Stojakovic [15] further proposed an algorithm to compute this feed-link in linear time.

For mobile ad hoc networks, Blazevic et al. [16] proposed so-called *terminode* routing to address holes in mobile network topologies. Durocher et al. [17] reviewed various geometric routing strategies for wireless network protocols.

Although some of these approaches share similarities with the techniques introduced in this article, they differ in the problem statement. In fact, to the best of our knowledge, we are unaware of any former study adding points to spatial systems based on geometric distances. Our problem setting has not been treated in terms of spatial databases that may store wide routing networks along with spatial representations of both the routing networks' vertices and POIs. We provide in this paper a conceptional foundation for describing routing networks globally on Earth's surface that is then translated to the field of spatial databases.

### 1.2. Structure of the Paper

More verbosely, we start with some preliminaries that introduce graphs and manifolds in Section 2. Briefly, we want to represent our routing network as a special class of graphs that can be mapped on a surface that is used for geometric reasoning. A modification of the routing network is conducted based on the geographic representation of the graph. Basically, we use manifolds for geometric reasoning. In order to connect the actual network with geometric points, we demand the routing network to be projectable on the manifold. This will allow us to do geometrical distance measuring. For that, we introduce a lower bound in Section 2.4 that we use to elaborate upon a theoretical solution in Section 3. The provided approach is a solution to the research hypothesis with respect to the fact that the geometrical shape of the manifold gets truthfully respected. Translation to spatial databases is conducted in Section 4. We further evaluate the implementation in Section 4.1 while providing an outlook in Section 4.3.

## 2. Preliminaries

Let us recollect some essential text-book concepts such as graphs, manifolds and metrics. To this end, we can formulate our problem in terms of this section.

### 2.1. Graphs

In order to represent a routing network on the Earth, let us recall the definition of a graph [18]:

**Definition 1.** *A directed weighted graph $G := (V, E, c)$ is a triple consisting of a vertex set $V$, an edge set $E \subseteq V \times V$ and a cost measure $c : E \to [0, \infty)$. Let us denote for two vertices $v_1, v_2 \in V$ with $(v_1, v_2) \in E$ the edge that connects $v_1$ to $v_2$. To avoid the excessive usage of brackets, we simplify the expression $c((u, v))$ to $c(u, v)$ for an edge $(u, v) \in E$. A walk $P := (v_1, \ldots, v_n)$ is a consecutive succession of vertices for an arbitrary $n \in \mathbb{N}$ such that there exists an edge $(v_i, v_{i+1}) \in E$ for each $i \in \{1, \ldots, n-1\}$. If the vertices $v_i$ and $v_j$ of a walk $P := (v_1, \ldots, v_n)$ are pairwise different for all $i \neq j$, we call $P$ a* path. *We say the walk $P$ with $n \in \mathbb{N}$ vertices follows "from $a \in V$ to $b \in V$" when $v_1 = a$ and $v_n = b$. We call $G$* connected *if there exists a walk from $a$ to $b$ for all $a, b \in V$. Moreover, we define the length of the walk $P = (v_1, \ldots, v_n)$ by $\ell(P) := \sum_{i=1}^{n-1} c(v_i, v_{i+1})$. Further, we denote with $\ell(a, b)$ the infimum of the lengths of all walks from $a$ to $b$, i.e.,*

$$\ell(a, b) := \inf\{\ell(P) : P \text{ walk from } a \text{ to } b\}.$$

We further want to examine a special class of graphs whose path length function $\ell$ is restricted in such a way that we can find a metric that is a lower bound of $\ell$. This restricts $\ell$ to be member of the following function class:

**Definition 2.** *Let $V$ be a set. A mapping $\ell : V \times V \to \mathbb{R}$ is called a* quasimetric *if it is a metric which does not need to fulfill the symmetry property.*

**Remark 2.** *For our approach, we neglect a possible symmetry property of routing networks. In fact, cost functions based on fuel or calorie consumption and estimated time are valid examples for directed networks that are in general non-symmetric.*

**Definition 3.** *If $E \neq \varnothing$ and $c$ of a connected graph $G := (V, E, c)$ supports the conditions*

$$\begin{aligned} (v, v) \in E \text{ for each } v \in V && \text{(reflexivity)}, \\ c(u, v) = 0 \text{ if and only if } u = v && \text{(definiteness)}, \end{aligned}$$

*then $\ell : V \times V \to \mathbb{R}$ is a quasimetric. We then call $G$ a* quasimetric network *and $\ell$ the quasimetric induced by $c$.*

See [19] for an indexing data structure built on a quasimetric network.

**Lemma 1.** *$\ell$ respects the triangle inequality, although $c$ is not required to hold this property.*

**Proof.** Because $c(e) \geq 0$ for all $e \in E$, a shortest walk will always be a simple path: Every walk that contains a circle is not a path. It can be made simple by removing all circles. However, this will also shorten the length of the walk. Hence, the definition of $\ell$ stays the same when taking the infimum of the length of all paths. We will call a walk that meets this condition a *shortest path*. Hence, we yield:

- From $\ell(P) \geq 0 \ \forall P$, we obtain the non-negativity of $\ell$.
- As $\ell(P) > 0$ for all walks from $u$ to $v \in V, u \neq v$, we obtain by definition $\ell(u, v) > 0$ for all $u, v \in V, u \neq v$. As $c(v, v) = 0$ for each $v \in V$, we can conclude that $\ell((v, v)) = 0$ for each $v \in V$. Thus, we yield the positive definiteness of $\ell$.
- If we define the concatenation of walks by $(v_1, \ldots, v_n) \circ (w_1, \ldots, w_n) :=$

$$\begin{cases} (v_1, \ldots, v_n, w_2, \ldots, w_n) & \text{if } v_n = w_1, \\ \text{undefined} & \text{otherwise}, \end{cases}$$

the triangle inequality is simple to show: For arbitrary $u, v, w \in V$, let $P_{uv}$ be a path from $u$ to $v$ and $P_{vw}$ a path from $v$ to $w$. Then, we can generate a walk $P_{uw} := P_{uv} \circ P_{vw}$ by combining both paths, so we have $\ell(P_{uw}) = \ell(P_{vw}) + \ell(P_{uv})$. Applying the infimum over all walks from $u$ to $w$ yields the triangle inequality.

$\square$

**Remark 3.** *In some technical scenarios, the definiteness is too restrictive. For instance, a database user may want to store two vertices with zero distance when one node shall represent the actual POI and the other the street segment at which the POI is located. If we drop definiteness in Definitions 2 and 3, we have to take care with the definition of the embedding i below, cf. Example 5.*

### 2.2. Manifolds

With regards to the vast number of encoding formats for spatial data, we describe our approach on a theoretical level, independently of any encoding. Therefore, we emphasize the concept of manifolds [20] in order to model space. The core idea is that we use several charts $\left\{ \varphi_j : V_j \to \mathbb{R}^2 \right\}_{j \in J}$ that map locally to a plane. The images of these charts can be glued to transit local reasoning over multiple planes. The following definition gives a precise characterization:

**Definition 4** ([20])**.** *A compact n-dimensional* manifold *is a compact topological space M for which a finite family of homomorphisms $\left\{ \varphi_j : V_j \to \mathbb{R}^n \right\}_{j \in J}$ exists with $V_j \subseteq M$ open for each $j \in J$ such that:*

- $\bigcup_{j \in J} V_j = M$.
- $\forall j, k \in J : V_j \cap V_k \neq \varnothing \Rightarrow \varphi_j \circ \varphi_k^{-1} : \varphi_k(V_k \cap V_j) \to \varphi_j(V_k \cap V_j)$ *is a homomorphism, called the* transition map.

  *Each $\varphi_i$ is called a* chart*, and $\left\{ \varphi_j : \varphi_j : V_j \to \mathbb{R}^n \right\}_{j \in J}$ is called an* atlas *of M.*

In differential geometry, the earth is often modeled as an oriented surface $M \subset \mathbb{R}^3$, i.e., a two-dimensional, topological manifold that embeds to the space $\mathbb{R}^3$ equipped with the Euclidean metric. We use the transition map in order to prolong geodesics over multiple maps. The $V_j$ are open, connected subsets of $M$ that cover $M$. For example, any sphere or spheroid is a surface and $\phi_1$ and $\phi_2$ are the charts of this manifold. Further, every oriented surface has a Gauss map [21] that defines the normal field $\nu : M \to \mathbb{R}^3 \subset S^2$. Informally, $M$ encodes the position on the surface, while $\nu$ allows us to express elevation relative to the surface.

### 2.3. Routing Networks

We now introduce another manifold $N$ that accounts for routing networks. In general, we will allow $N \not\subset M$, i.e., bridges, tunnels or other partially inaccessible street segments can be expressed by this model.

**Definition 5.** *Let $(V, E, c)$ be a quasimetric network. Further, let $i : V \hookrightarrow N, E \to \mathbb{P}(N)$ be an embedding in a manifold $N \subset \mathbb{R}^3$, where $\mathbb{P}(N)$ is the power set of N. Let the following conditions hold:*

- $i \mid_V : V \hookrightarrow N \subset \mathbb{R}^3$ *with $\bigcup_{e \in E} i(e) = N$;*
- *For each $n \in N$, there exists an $m \in M$ and an $h \in \mathbb{R}$ such that $m + h\nu(m) = n$ (embedding property). This property is based on the fact that street segments may not be on the surface. Informally, the parameter h counts for the altitude difference between the ground and street; e.g., bridges have positive heights and tunnels negative heights. This separation between spherical data and altitude is also common when dealing with the WGS84 format that encodes points on M by latitude and longitude [22].*
- *Let $\pi_M \circ i : V \to M$ be the (canonical) projection of the graph onto M. Then, we assume that for all $e \in E$ there exists some $j \in J$ with $\pi_M \circ i(e) \subset V_j$, i.e., we stipulate that the complete image of each edge is contained in at least one chart's image (containment property).*
- *For each edge $e = (a, b) \in E$, there is a line $\gamma_e : [0, 1] \to N$ with $\gamma_e(0) = i(a), \gamma_e(1) = i(b)$, and $i(e) = Im \, \gamma_e$, i.e., $\gamma_e$ is a linear continuous, parameterized, geodesic line with a and b as endpoints (line-string property).*

- *Two of these lines $\gamma_1$ and $\gamma_2$ are called equivalent when $\gamma_1 = \gamma_2$ or $\gamma_1(1 - \cdot) = \gamma_2$. If we denote equivalence with $\sim$ and define the quotient set $\Gamma := \{\gamma_e : e \in E\}/\sim$, then $\{\gamma(0,1) : \gamma \in \Gamma\}$ shall be a disjoint partition of $N \setminus i(V)$ (disjoint property).*

  *We then call the tuple $G = (V, E, c, i, N)$ a routing network. $i(V) \subset N$, since $G$ is connected. In particular, if there exists an injection $N \hookrightarrow V_j \subset \mathbb{R}^2$ for some $j \in J$, then the routing network $G$ is planar. Let us use the symbol $\mathcal{G}$ for the class of routing networks.*

**Remark 4.** *The equivalence relation $\Gamma$ can be understood as a lift of an equivalence relation that renders a directed graph undirected by*

$$E/\sim \ := \{\{u, v\} : (u, v) \in E \text{ or } (v, u) \in E\}.$$

Let us consider the scenario of a spatial recommender system that uses the current location of a user to search for close POIs. The geolocation is collected by a mobile navigation system. Hence, the recommender has to map this polled location to its routing network in order to compute distances. The query point might not even be in the image of *i*. We have to add these points to the graph by enhancing the number of vertices and edges in such a way that these particular points can be accessed by our network, as well. A vertex on which a geographic point is mapped is called a *location reference*. We will show some ideas on how to model location references so that shortest path evaluation should retrieve good results in relation to real-world expectation.

*2.4. Lower-Bounding Metrics*

The cost function *c* of a routing network can express various measures: distance length, travel time, fuel consumption, etc. The definiteness of *c* does not allow us to add new vertices to networks with edges of zero cost. In fact, we want to give the new reference location a penalty for the expense to travel from (to) the current location to (from) the routing network. Most scenarios inherit a lower bound for this expense, e.g., the beeline when *c* encodes travel distances. We discuss some further scenarios and show possible lower bounds, which we formalize in the following definition:

**Definition 6.** *We call a metric $d : N \cup M \times N \cup M \to \mathbb{R}$ such that $d(i(a), i(b)) \leq \ell(a, b)$ for all vertices $a, b \in V$, where $\ell$ denotes the quasimetric induced by c, a lower-bounding metric of $\ell$. For a set $U \subseteq N$ and a point $n \in N$, we use the common shortcut $d(U, n) := \inf_{u \in U} d(u, n)$ with $d(\emptyset, n) = \infty$.*

**Example 1.** *Let $G = (V, E, c, i, N)$ be a routing network in which $i(e)$ shall represent a street segment for an arbitrary $e \in E$. For each street segment e, the network can query with $c(e)$ the estimated time needed to get from the start point of e to its end point. The induced quasimetric $\ell$ respects the triangle inequality: For each start and end point $s, t \in V$, the obtained value of $\ell(s, t)$ is the length of a shortest path from s to t, i.e., a path whose accumulated, estimated time is shortest.*

**Example 2.** *Let $c(u, v)$ be the Euclidean distance between the mappings of $i(u)$ and $i(v)$ to a local plane. Such a mapping exists for each edge due to the containment property. For such a c, the following metrics are valid lower bounds:*

- *A common local lower-bounding metric is the beeline when the network is mapped to a local plane. It would be tedious to adhere multiple charts by the transition map in order to calculate the beeline between two distant points. Fortunately, it is easy to calculate the geodesic on M by approximation [23]. For example, Vincenty's algorithm [24] is a good approximation for calculating geodesics between two different points on the surface of the Earth.*
- *The Euclidean metric in $\mathbb{R}^3$ is a lower-bounding metric. In particular, this metric is also a lower bound of the local beeline because it is allowed to neglect the curvature of the manifold M, cf. Figure 1.*

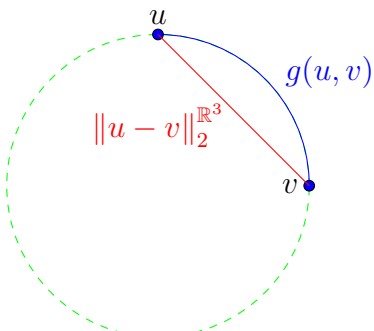

**Figure 1.** Curvature on an ellipsoid embedded in $\mathbb{R}^3$. The ellipsoid's surface is visualized in dashed green color. We take two points $u$ and $v$ on this ellipsoid. If we cut the ellipsoid along these two points, we obtain a circle on which both points lie. The arc (blue color) that connects both points is a geodesic $g(u,v)$ of the surface [25]. A straight line (red color) would be shorter than the geodesic and hence the $l^2$-norm $\|\cdot\|_2^{\mathbb{R}^3}$ is a lower bound.

**Definition 7.** *Let $G = (V, E, c, i, N)$ be a routing network and $d$ a metric. If $d$ is a lower-bounding metric of $\ell$, where $\ell$ is the quasimetric induced by $c$, we say that $G$ respects the metric $d$.*

### 3. Linkage to Network

Let us again formulate our posed problem on a surface $M$: Given a routing network $G = (V, E, c, i, N) \in \mathcal{G}$, its geometric representation $N$ and a lower-bounding metric $d$, we want to add a new vertex $v' \notin V$ to $G$ in such a way that:

- The resulting routing network stays valid. In particular, the graph shall remain connected, i.e., there is an edge with $v'$ as an end.
- The modified geometric representation $i'$ maps $i'(v')$ to $M$.
- $i(v') \in M$. Informally, this means that a pedestrian can access $v'$ from the ground.
- $d$ is still a valid lower-bounding metric.

We call this transformation a *linkage* and pose now a formal definition of it:

**Definition 8.** *Let $G = (V, E, c, i, N) \in \mathcal{G}$ and $\ell$ be the quasimetric induced by $i$. A linkage is some mapping $\sigma : M \times \mathcal{G} \to \mathcal{G}$ with the property that the resulting quasimetric network $G' := (V', E', c', i', N') := \sigma(a, G)$ holds the conditions:*

- *$i' : E' \cup V' \to N'$ with $i'|_V = i|_V$ and $i'|_{(E \cap E')} = i|_{(E \cap E')}$.*
- *$V' = V \cup \{v'\}$ with $i'(v') = a$ for some $v'$ (that is not necessarily part of $V$).*
- *$i'(V' \cup E') \subset N'$; hence, $a \in N' \cap M$.*
- *There exist some $x, y \in V$ with $(v', x) \in E'$ and $(y, v') \in E'$.*
- *$c' : E' \to \mathbb{R}$ with $c'|_{E \cap E'} = c$.*
- *$G'$ respects the metric $d$.*

*Hence, $\ell(x, y) = \ell'(x, y)$ for all vertices $x, y \in V$ where $\ell$ and $\ell'$ are the quasimetrics induced by $i$ and $i'$, respectively.*

The crux of this problem lies in the construction of appropriate edges or vertices that sustain the properties of a routing network. In the following, we introduce some construction steps to elaborate a simple but complete solution in the end. First, we treat the special case that we want to add some point $a \in M \cap N$ that belongs to $G$, i.e., the location reference $v \in V$ with $i(v) = a$ already exists. Hence, nothing has to be done. Second, $a \in M \cap N$ may be in the image $i(e)$ of some edge $e \in E$. Then, we split $e$ into two pieces by linear interpolation and add the location reference of $a$ as an intermediate piece. Formally, we have:

**Example 3.** *We define the* linear interpolation *$G' := \sigma(a, G)$ of $G := (V, E, c, i) \in \mathcal{G}$ for a point $a \in M$ as follows:*

1.  *If there exists some $v \in V$ such that $i(v) = a$, then $G' := G$.*

2. *If there exists an edge $e \in E$ such that $a \in i(e) \setminus i(V)$, then there is some $\lambda \in (0,1)$ such that $\gamma_e(\lambda) = a$, where $\gamma_e$ is the line induced by $i(e)$ due to the line-string property. Let $x, y \in V$ be the vertices connected by $e$ such that $e = (x, y)$. We add a new vertex $v' \notin V$ with $i(v') = a$. Let us set $c'(x, v') := \lambda c(x, y)$ and $c'(v', y) := (1 - \lambda)c(x, y)$. If $(y, x) \in E$, we further set, due to $\mathrm{Im}\ \gamma_{(x,y)} = \mathrm{Im}\ \gamma_{(y,x)}$, $c'(v', x) := \overleftarrow{\lambda} c(y, x)$ and $c'(y, v') := (1 - \overleftarrow{\lambda})c(y, x)$ with $\overleftarrow{\lambda} \in (0,1)$ such that $\gamma_{(y,x)}(\overleftarrow{\lambda}) = a$. In the end, we define $E'$ as the set $E$ without $\{(x, y), (y, x)\}$, but with the new edges used above - $\{(x, v'), (v', y)\}$, and $\{(v', x), (y, v')\}$ if $(y, x) \in E$.*

   *Let $\ell$ and $\ell'$ be the quasimetrics that are induced by $c$ and $c'$, respectively. Then, $\ell'\ |_{V \times V} = \ell$ holds due to $\ell(x, y) = c(x, y) = c'(x, v') + c'(v', y) = \ell'(x, y)$. Thus, we yield the beeline property with $i'(x, v') = \gamma_e([0, \lambda]), i'(v', y) = \gamma_e([\lambda, 1])$. With the same arguments, we are able to preserve the properties of a routing network for $(y, x)$, if $(y, x) \in E$. Because we have not changed the image of $i$, we can just set $N' := N$ and hence we are finished.*

3. *Otherwise, we are certain that $a \notin N$. We use a not yet defined method to create a new network $G' = (V', E', c', i', N')$ with some $E', c'$ and $N'$.*

   *In both specified cases, the linear interpolation is a linkage, i.e., $\sigma(G) := (V', E', c', i', N)$ fulfills the properties of Definition 8. See Figure 2 for an illustration.*

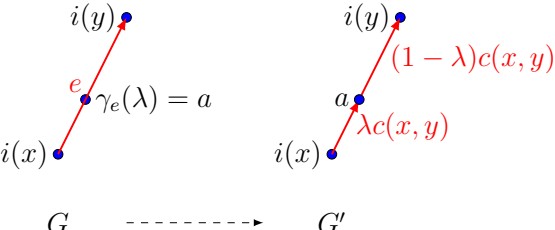

**Figure 2.** Linking a point $a$ by linear interpolation of an edge $e$.

Note that Example 3 takes care just for points that are already part of $N$. In the following, we try to fill case (3). Here, we can be sure that the point $a \in M$ to add does not belong to the image of $i$; hence, $a \notin N$. In order to keep the graph connected, we have to add some edges after inserting the location reference of $a$. For the search of suitable access points, we take only those segments of the routing network into account that intersect with the manifold $M$. Informally, the intersection takes account for accessibility from the ground. For example, we consider the internal segments of a tunnel ("below $M$") or bridge ("above $M$") inaccessible. To access a tunnel or bridge, a path to the entrance of a tunnel or the vertex that connects the bridge with $M$ has to be found. For a query point $a \in \varphi_j V_j$, if the routing network can be drawn on the surface, i.e., the condition $\pi_{V_j} N = N \cap V_j$ holds, then we do not have to care about inaccessible points.

The first example tries to solve the problem by simply adding an edge to the closest accessible vertex of the routing network, see Figure 3 for an example. Unfortunately, the resulting graph may not be a routing network anymore. Nevertheless, we can use this approach to elaborate upon Example 4.

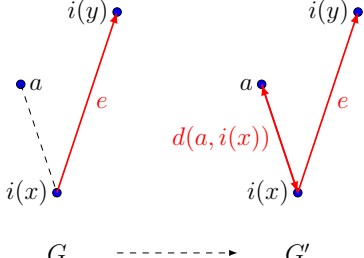

**Figure 3.** Vertex linkage. We link a point $a$ to the node $i(x)$ of the routing network.

**Counter-Example 1.** *We define the* vertex linkage *$G' := \sigma(a, G)$ of $G := (V, E, c, i, N) \in \mathcal{G}$ for a point $a \in M$ that is based on linear interpolation. Case (3) of Example 3 is implemented as follows: Define $V' := V \cup \{v'\}$ with some $v' \notin V$, and set $i'(v') := a$ and $i' \mid_V = i$ otherwise. Further, let $u := \operatorname{argmin}_{v \in V : i(v) \in M} d(i(v), a)$. We then define $E' := E \cup \{(u, v'), (v', u)\}$ with $c'(u, v') = c'(v', u) = d(a, i(u))$. In general, $\sigma(G) := (V', E', c', i', N \cup i'(u, v'))$ is not a valid linkage with regards to Definition 8. The problem arises when there exists an edge $e \in E$ with $i(e) \cap i(u, v') \neq \varnothing$. This is possible because there might exist an edge $e =: (x, y)$ with the following properties (cf. Figure 4):*

- *$d(i(e), a) < d(i(u), a)$ and hence $d(i(e), a) \leq \min\{d(i(x), a), d(i(y), a)\}$;*
- *There exist $\alpha < 0, w \in \dot{\gamma}_e^\perp(0)$ and $\lambda, \mu \in [0, 1]$ such that*

$$\gamma_e(\lambda) + w = i(v) \text{ and } \gamma_e(\mu) + \alpha w = a,$$

*where $\gamma_e$ is the geodesic of $e$ induced by $i$.*

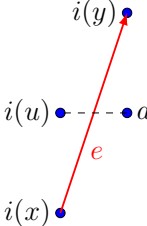

**Figure 4.** Vertex linkage creating a new edge intersecting with $e$. After this operation, the graph is no longer planar, a setting we want to avoid.

Instead of the closest vertex, we can also take some route segment as an access point; we split an edge like the linear interpolation of Example 3 in such a way that the intermediate piece $\tilde{v}$ acts as the new reference location. The image $i(e)$ of a suitable edge $e$ should have a point $b = i'(\tilde{v}) \in i(e) \cap M$ that is close to $a$. On a plane $\varphi_j V_j$ with $\pi_{V_j} N = N \cap V_j$, the point $b$ is determined by the perpendicular from $a$ to the closest edge with respect to $a$. In the general case, we have to follow the shortest geodesic from $a$ to the routing network. This approach is visualized in Figure 5.

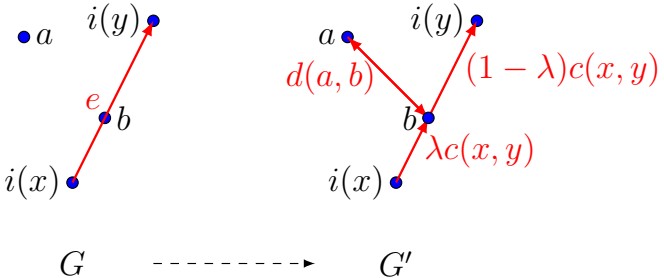

**Figure 5.** Edge Linkage. We link a point $a$ to a newly created node $b$ splitting the former edge $e$ of the routing network.

**Example 4** (Edge Linkage). *We define the* edge linkage *$G' := \sigma(a, G)$ of $G := (V, E, c, i, N) \in \mathcal{G}$ for a point $a \in M$ that is based on linear interpolation and implement the last procedure as follows: First, search for an*

$$e := \operatorname{argmin}_{e \in E} d(i(e) \cap M, a)$$

*and set $\lambda := \operatorname{argmin}_{\lambda \in [0,1]} d(\gamma_e(\lambda), a)$, where $\gamma_e$ is the geodesic line induced by $i(e)$. Let $x, y \in V$ be the vertices attached by $e$ such that $e = (x, y)$ and $\delta := d(\gamma_e(\lambda), a)$. There are now two cases to consider:*

1. $\lambda = 0$ or $\lambda = 1$, *i.e.*, $d(i(e), a) = d(i(x), a)$ or $d(i(e), a) = d(i(y), a)$. *Without loss of generality, let the equation $d(i(e), a) = d(x, a)$ hold. Now we can apply a vertex linkage on G by Counter-Example 1 (setting $u \leftarrow x$). This linkage holds the disjoint property.*

2. *Otherwise, $\lambda \in (0,1)$. Hence, there exists some $b \in N$ with $b \in i(e)$ and $d(i(e), a) = d(b, a)$. We apply linear interpolation on G with b and yield a new network $\sigma(b, G) =: \tilde{G} := (\tilde{V}, \tilde{E}, \tilde{c}, \tilde{i}, N)$ with $\tilde{i}(\tilde{v}) = b$ for some $\tilde{v} \in \tilde{V}$. Note that the rule applied by linear interpolation will not modify N. If we exchange G with $\tilde{G}$, the first case holds, since there exists an edge $\tilde{e} := \arg\min_{\tilde{v} \in \tilde{E}} d(\tilde{i}(\tilde{v}), a)$ with $\min_{\tilde{\lambda} \in [0,1]} d(\gamma_{\tilde{e}}(\tilde{\lambda}), a) \in \{0, 1\}$.*

**Proof.** For (2), we need to show that there is no $e' \in E'$ that intersects with $E' \setminus \{e'\}$. Let us assume that such an $e' \in E'$ exists. As we have merely exchanged $e$ with $\{(x, v), (v, y)\}$ in $E$, $e'$ has to intersect either $(x, v')$ or $(v', y)$. Then, $d(i(e'), a) < \min\{d(i(x, v'), a), d(i(v', y), a)\} = d(i(e), a)$, a contradiction to the selection of $e = (x, y)$ to be the closest edge to $a$. □

Retrieving the closest edge/vertex is usually conducted by a nearest-neighbor search on the database index. A common spatial index structure is the R-tree [26] that builds bounding boxes of geometries. For instance, Roussopoulos et al. [27] elaborated upon a branch-and-bound algorithm that evaluates distances between bounding boxes.

When used in real-world scenarios, it is often the case that some point $a \in M$ is very close to a mapped vertex, but the coordinates do not match exactly. More precisely, for an $\epsilon > 0$ small enough, we have $\min_{v \in V} d(i(v), a) < \epsilon$. Let us consider that we have a huge collection of sensor-collected geolocations that shall be linked to an existing routing network. For the same location, measured values tend to have small differences. Hence, it might be advisable for practical reasons to condense very close points to one vertex. If we redefine the properties of $i$ in such a way that we have $i(v) \subset N$ instead of $i(v) \in N$, then we could do the following trivial optimization:

**Example 5** (Fuzzy matching). *If there exists some $v \in V$ with $\min_{v \in V} d(i(v), a) < r$, then add $a$ to $i(v)$, i.e., $i' |_{V \setminus \{v\}} = i$ and $i'(v) = i(v) \cup \{a\}$. We define the $r$-snap $\sigma_r : M \times \mathcal{G} \to \mathcal{G}$ with a threshold $0 \leq r < \infty$ as a modification of our linear interpolation approach (Example 3), which makes the above rule its highest priority. Choosing the right r is situation-dependent, e.g., a small r would be more preferable in network-dense areas. Note that this approach resembles "snapping" in computer graphics.*

To recap, we first studied linear interpolation as a method to add a point $a$ to the routing network for the special case that $a$ is on an already-existing edge, which we split at $a$. Next, we studied ways in how to link $a$ when $a$ is on no existing edge of the routing network. There, we observed that the vertex linkage linking $a$ to the closest vertex does not retain planarity in general. Luckily, we could show that the edge linkage retains this property, making it a suitable candidate for tackling our linkage problem. Finally, we introduced the $r$-snap technique, which allows us to apply linear interpolation for close enough points while sacrificing accuracy. In what follows, we present an implementation of the edge linkage approach on a database system.

## 4. Implementation

The key extension of SQL with respect to geoscience is the technical report "Simple Features for SQL" of the Open Geospatial Consortium (OGC) [4,28]. Common databases already offer an addition to its core framework to address this concept. For PostgreSQL, there exists the extension PostGIS [29] that follows OGC's described standard. Hence, we describe our implementation in terms of the OGC standard.

While the extensions pgRouting and PostGIS are written in C/C++, our middleware was implemented in Java. The linkage and routing queries were conducted by JDBC commands with OGC's SQL extension. By restricting ourselves to JDBC calls, our solution was independent of a specific relational database management system, as long as OGC's spatial extensions to SQL are provided.

We have already employed the implementation described below in one demonstration [30]. There, the end user could mark arbitrary locations as favorable points by simple mouse-clicks on a map overlay. According to the generated, geometric objects, a database query on a routing network was issued. Because the user was not restricted to selecting points on the routing network, a linkage had to be performed. For this scenario, we used the OSM map of Munich (of the year 2013) that we imported statically into the PostgreSQL database. Access to the routing network was obtained by the additional extension pgRouting [31], which sits on top of PostGIS. Figure 6 depicts the software layers of our employed solution. Note that there are also other frameworks with similar concepts, cf. [32]. The translation of our linkages to the language of OGC was performed easily:

- $d$ was given by `ST_Distance`.
- $\ell$ was computed by a shortest path algorithm on pgRouting's network. Common algorithms such as Dijkstra and A* were available.
- `ST_ShortestLine` represented the geodesic function $g$.
- $\text{argmin}_{a \in A}\, d(a, v)$ with $A \subseteq E$ or $A \subseteq V$: We used `ST_DWithin` as a pre-filter for distant edges/vertices before the exact distances to all remaining edges/vertices were calculated.
- The split of an edge $(u, v)$ into $(u, v')$ and $(v', u)$ with $i(v') \in i(u, v)$ was performed by calling `ST_Line_Locate_Point` and then generating both new edges with two calls of `ST_Line_Substring`.

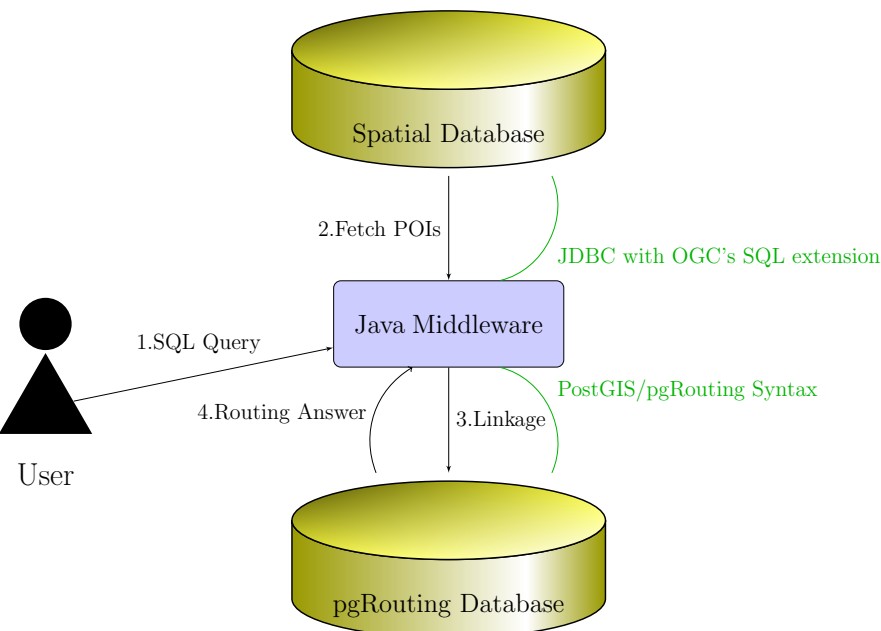

**Figure 6.** Architecture design chart. Our Java middleware parses a query from a user, fetches POIs from a spatial database, amends temporarily the routing network and finally runs a routing query on the pgRouting database.

On a query, the network would be modified in such a way that the user's specified geolocations were represented in the routing network. After the query was evaluated, we restored the network back to its initial state.

### 4.1. Evaluation

We evaluated the edge linkage as part of our demonstration [30]. Our evaluation was conducted on top of our middleware in Java using JDBC as a connection (cf. Figure 6). 100 to 800 points on the map were randomly marked for linkage. After each point was linked to the routing network, distances to a fixed set of POIs were calculated by pgRouting's Dijkstra implementation. The execution times of Table 1 were gathered on a single Debian

6.0 node with an Intel(R) Xeon(R) CPU E5540. For each geometric point a user specified, the framework linked this point to the routing network and calculated the distances to some fixed POIs. It is easy to see that the routing with Dijkstra was by far more costly than the linkage.

**Table 1.** Running time for linkage and routing on the setting described in Section 4.1. The ratio of execution time between linkage and routing is almost constant with variance in the number of nodes $|V|$.

| $|V|$ | Linkage | Routing | Ratio |
|---|---|---|---|
| 100 | 0.40 s | 8.06 s | 4.96% |
| 200 | 0.74 s | 15.21 s | 4.87% |
| 400 | 1.33 s | 32.26 s | 4.12% |
| 800 | 2.47 s | 59.69 s | 4.14% |

*4.2. Expectations on Larger Datasets*

The naive Dijkstra implementation runs in $O(|E| + |V| \lg |V|)$ time [33,34]. Since its running time super-linearly depends on the number of vertices, its performance deteriorates when scaling up the number of vertices. Here, indexing data structures for routing networks have been proposed, such as hierarchical hub labels [35], contraction hierarchies [36–39] or a combination of those [40,41]. To adapt our approach for such an indexed routing network, we not only have to perform the linkage on the plain routing network, but we must also update the underlying indexing data structure, which is used for the shortest path query. Updating hub labels [42] and contraction hierarchies [43] have been studied, but it is unclear whether we can put the update time in relation to a shortest path query.

*4.3. Outlook*

One may argue that the here-proposed edge linkage is still a naive implementation because it neglects possible detours. Fortunately, by the introduction of both vertex and edge linkage, it is straight-forward to construct a valid linkage that creates a feed-link [14] from the non-connected point. Another complaint may arise that the complete use of $M$ is quite naive. Let us imagine that the edge created to link an off-road point to the routing network leads over a river, lake, mountain gorge, etc. Pedestrians or travelers with common means of transportation are not able to overcome these obstacles and have to seek an alternate route. Real datasets such as OSM represent obstacles by a polygon in which the actual obstacle is contained, such as in Figure 7. If we stipulate that every chart of $M$ is large enough such that every obstacle can be rendered in at least one chart's image, any lower-bounding metric can be modified to respect the presence of obstacles. The additional computation can be conducted, for example, by Hershberger and Suri's algorithm [44] which solves the shortest path problem on a plane with obstacles. Lastly, we could adapt our introduced concept to more detailed routing networks. New models enrich network information, for instance, with affordance [45] to describe possible physical actions.

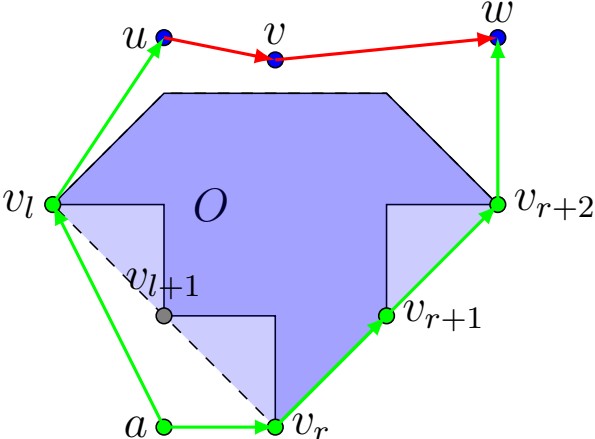

**Figure 7.** Linking point *a* to a routing network with the beeline crossing an obstacle *O* (dark blue). We consider the vertices on the convex hull of *O* and start creating a path (green arrows) from *a* to along the vertices on the convex hull of *O* to the routing network, where we find *u* and *w* to be the two closest nodes to *a* with respect to the Euclidean distance.

## 5. Conclusions

Mixing graph theory with geospatial information is a still-emerging topic that gives answers to interesting, novel questions. We have reasoned about several approaches on how to add off-road points to a routing network. A vertex linkage has the simple advantage that it is easy to implement and in most cases offers the desirable outcome. An edge linkage, on the other hand, is more fine-grained, because its constructed segment often is shorter than the new edge of the vertex linkage. Besides the fact that it is more time-consuming than a simple vertex linkage, it additionally exchanges an edge with a constructed vertical. Without the knowledge of additional information, such as terrain, unrecorded new street segments or temporary construction works, it goes without saying that our proposed approaches are far away from being optimal solutions.

**Funding:** This research was funded by JSPS KAKENHI with grant numbers `JP21K17701` and `JP21H05847`.

**Institutional Review Board Statement:** Not applicable.

**Informed Consent Statement:** Not applicable.

**Data Availability Statement:** The used OSM map from the Munich area can be directly downloaded from http://api.openstreetmap.org/api/0.6/map?bbox=11.54,48.14,11.543,48.145 (accessed on 20 April 2022).

**Acknowledgments:** The author is grateful to Roland Glück for insightful discussions about this topic.

**Conflicts of Interest:** The authors declare no conflict of interest.

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
