# Peer review of "Linking Off-Road Points to Routing Networks"

_algorithms, doi:10.3390/a15050163_

Round 1
Reviewer 1 Report
- Results: Recommend to be Major revisions
This paper bridges the gap between routing networks and spatial geometry by a global matching of geometric points to routing networks. Although graph theory has already been introduced to spatial reasoning, current spatial database systems do not provide out-of-the-box routing on geometric points that are not matched on the graph. Methods that connect new reference locations to the graph render different routing results. Moreover, current solutions break reasoning down to local analysis.
This paper is with minor merits for Algorithms, i.e., poor writing skills and lacking of insight analysis, it requires major revisions.
Firstly, for Section 1, authors should provide the comments of the cited papers after introducing each relevant work. What readers require is, by convinced literature review, to understand the clear thinking/consideration why the proposed approach can reach more convinced results. This is the very contribution from authors. In addition, authors also should provide more sufficient critical literature review to indicate the drawbacks of existed approaches, then, well define the main stream of research direction, how did those previous studies perform? Employ which methodologies? Which problem still requires to be solved? Why is the proposed approach suitable to be used to solve the critical problem? We need more convinced literature reviews to indicate clearly the state-of-the-art development.
For Section 2, authors should introduce their proposed research framework more effective, i.e., some essential brief explanation vis-à-vis the text with a total research flowchart or framework diagram for each proposed algorithm to indicate how these employed models are working to receive the experimental results. It is difficult to understand how the proposed approaches are working.
For Sections 3 and 4, authors should use more alternative models as the benchmarking models, authors should also conduct some statistical test to ensure the superiority of the proposed approach, i.e., how could authors ensure that their results are superior to others? Meanwhile, authors also have to provide some insight discussion of the results.
Reviewer 2 Report
The article is well written, logically structured, and addresses an important practical problem in improving current spatial database systems. The author points out some shortcomings of common spatial systems, such as non-planar network graphs or very distant points whose projection on a compact plane cannot take angles and distances into account simultaneously. Therefore, he proposes to improve a global routing network in such a way as to link the following two issues: queries based on geometric information and matching the geometric position with the network (the so-called problem 1). It is therefore necessary to add additional vertices to a given routing network's graph to represent new geometric points. Connecting the actual network with the geometric points provides a projection of the routing network onto a topological manifold.
In terms of content, the paper fits the scientific profile of the journal Algorithms. For the paper to be published, the author should address the following issues:
- The proposed solution has only been implemented on a local scale, as only the 2013 map of Munich has been considered. However, the Author's ambitions are much higher, as he is interested in very distant points and geodesic lines. To what extent will the complexity of the geolocation system increase if a global implementation is done?
- The next comment is about problem 1: If we have only one problem to solve then we don't need to number it. I advise changing the form of the issue raised from "problem" to "research hypothesis".
- The author writes about his proposed solutions using the plural "we", which appears exactly 111 times in the 13-page article. Does anyone else co-author the article?
Round 2
Reviewer 1 Report
Authors have completely addressed all my concerns.